# Terminator-free template-independent enzymatic DNA synthesis for digital information storage

Henry H. Lee [1,2,4], Reza Kalhor [1,2,4], Naveen Goela [3,4], Jean Bolot[3] & George M. Church [1,2]

DNA is an emerging medium for digital data and its adoption can be accelerated by synthesis processes specialized for storage applications. Here, we describe a de novo enzymatic synthesis strategy designed for data storage which harnesses the template-independent polymerase terminal deoxynucleotidyl transferase (TdT) in kinetically controlled conditions. Information is stored in transitions between non-identical nucleotides of DNA strands. To produce strands representing user-defined content, nucleotide substrates are added iteratively, yielding short homopolymeric extensions whose lengths are controlled by apyrase-mediated substrate degradation. With this scheme, we synthesize DNA strands carrying 144 bits, including addressing, and demonstrate retrieval with streaming nanopore sequencing. We further devise a digital codec to reduce requirements for synthesis accuracy and sequencing coverage, and experimentally show robust data retrieval from imperfectly synthesized strands. This work provides distributive enzymatic synthesis and information-theoretic approaches to advance digital information storage in DNA.

[1] Department of Genetics, Harvard Medical School, Boston 02115 MA, USA. [2] Wyss Institute for Biologically Inspired Engineering at Harvard University, Boston 02115 MA, USA. [3] Technicolor Research & Innovation Lab, Palo Alto 94306 CA, USA. [4]These authors contributed equally: Henry H. Lee, Reza Kalhor, Naveen Goela. Correspondence and requests for materials should be addressed to H.H.L. (email: hhlee@genetics.med.harvard.edu) or to G.M.C. (email: gchurch@genetics.med.harvard.edu)

D NA is a compelling data storage medium given its superior density, stability, energy-efficiency, longevity, and lack of foreseeable technical obsolescence compared with commonly used electronic media[1,2]. Recent studies have demonstrated that digital data can be written in DNA, stored, and accurately read[3–9]. To date, DNA for information storage has been produced by phosphoramidite chemistry, a powerful method that has matured over several decades[10] for synthesizing synthetic DNA with single-base accuracy to drive biological research[11]. However, this organic synthesis method can limit the quality and quantity of synthesized DNA owing to depurination[12], acetonitrile availability[13,14], and price[3,4,6]. As a result, there is a renewed interest in developing enzymatic approaches to DNA synthesis, which can occur in aqueous conditions and yield longer DNA products with reduced reagent costs[15–17].

Although polymerases naturally synthesize DNA, their use for de novo production of customized sequences is still in its nascency. Enzymatic DNA synthesis strategies have been described that use protected nucleotide analogs and/or engineered polymerases to synthesize DNA with a precise sequence[18–21]. Information storage, however, does not require DNA with single-base precision or accuracy. Here, we describe a kinetically controlled system that uses TdT to catalyze the linkage of naturally occurring nucleoside triphosphates (dNTPs) to synthesize customized DNA strands with short homopolymeric extensions. As a result, we encode information in transitions between non-identical nucleotides, rather than in each nucleotide. We also devise and demonstrate a codec to support accurate data retrieval from imperfectly synthesized strands and mathematically evaluate the parameters affecting the storage capacity of this system.

## Results

**Enzymatic DNA synthesis.** Terminal deoxynucleotidyl transferase (TdT) is a template-independent DNA polymerase that adds all four dNTPs to the 3′ termini of DNA strands[15,22–25]. In order to utilize TdT for de novo synthesis, a strategy for controlling polymerization is required. Inspired by previous work[26–28], we decided to leverage apyrase, which degrades nucleoside triphosphates into their TdT-inactive diphosphate and monophosphate precursors. Apyrase limits DNA polymerization by competing with TdT for nucleoside triphosphates. We first optimized a mixture containing a tuned ratio of these two enzymes such that a nucleoside triphosphate is added at least once to each strand by TdT before being degraded by apyrase (Supplementary Fig. 2, Supplementary Note 1). We then characterized polymerization activity as a function of various buffer conditions, additives and divalent cations, enzyme to initiator ratio, and nucleoside triphosphate concentrations (Supplementary Figs. 3–5, Supplementary Note 1). We found that although the addition of cobalt results in longer strands, they are more heterogeneous in length (Supplementary Fig. 3). Importantly, we also found that the terminal base at the 3′-end of the initiator has a significant effect on the nucleotide concentration required for extension, and that TdT prefers initiators that end in purines to those ending in pyrimidines (Supplementary Fig. 6, Supplementary Note 1). Accordingly, we determined the lowest required concentration for each nucleoside triphosphate that would result in extension of the initiator regardless of its terminal 3′ base. The combination of these characterizations and optimizations yields a system where the addition of a series of nucleotides results in stepwise increases in the length of synthesized DNA (Fig. 1, Supplementary Fig. 7).

The synthesis system consists of a mixture of TdT, apyrase, and short oligonucleotide initiators (Fig. 1a, Supplementary Fig. 1B). Upon addition of a nucleoside triphosphate substrate, TdT extends the initiators until all added substrate is degraded by

apyrase. We define the number of polymerized nucleotides as "extension length". Subsequent nucleoside triphosphates are added to continue the synthesis process. Although the extension length for each added nucleoside triphosphate may vary, the resulting population of synthesized strands all share the same number and sequence of nucleotide transitions (Fig. 1b).

These transitions between non-identical nucleotides encode user-defined information (Fig. 1c). Given three possible transitions for each nucleotide, we use trits, a ternary instead of binary representation of information, to maximize information capacity. To convert information to DNA, information in trits is mapped to a template sequence that represents the corresponding transitions between non-identical nucleotides starting with the last nucleotide of the initiator. Enzymatic DNA synthesis of each template sequence produces "raw strands", or strands[R], which can be physically stored. To retrieve information stored in DNA, strands[R] are sequenced and transitions between non-identical nucleotides extracted, resulting in "compressed strands", or strands[C]. If a strand[C] is equivalent to the template sequence, the strand (compressed or raw) is considered "perfect" and the information is retrieved by mapping its sequence of transitions between non-identical nucleotides back to trits.

To demonstrate the storage of information, we encoded and synthesized "hello world!", a message containing 96-bits of ASCII data (Fig. 2a). We split this message into twelve individual 8-bit characters and prefixed each character's bit representation with a 4-bit address to denote its order. These 144 total bits of information, including addressing, were expressed in trits and mapped to nucleotide transitions (Fig. 1c), resulting in 12 eight-nucleotide template sequences (Supplementary Table 2). We then synthesized all twelve template sequences (H01–H12) in parallel on bead-conjugated initiators while washing every two cycles. Following the last synthesis cycle, all strands[R] were ligated to a universal adapter, PCR amplified, and stored as a single pool (Methods).

**Data retrieval and error analysis.** We used Illumina sequencing to read out the synthesized strands[R] and assess the information stored in corresponding strands[C] (Methods). We started by analyzing the perfect strands. We found that the extension length for each nucleotide varied based on the type of transition (Fig. 2b, Supplementary Fig. 9, Supplementary Table 3). As a result, perfectly synthesized strands for each template sequence may be of variable raw length. In addition, when extension lengths were compiled for each nucleotide across strands and positions based on type of transition, we observed that these lengths were qualitatively consistent between bead-conjugated (Fig. 2c) and freely diffusing initiators (Supplementary Fig. 6, Supplementary Note 1). For example, the median extension lengths of C when following A, T, or G were among the lowest. Conversely, the median extension lengths for A, T, and G when following C were among the highest. Considering all synthesized strands, we found stepwise increases in the median raw lengths with an increasing number of non-identical nucleotides (compressed strand length), indicating controlled polymerization for the population of strands over multiple cycles (Fig. 2d). However, compared with a median length of 30 nucleotides for all perfect strands[R], the median length for all synthesized strands[R] was 26 bases, suggesting that not every strand polymerized the added nucleoside triphosphate in each cycle (Fig. 2e, Supplementary Fig. 10).

To identify the types and magnitude of synthesis errors, we aligned all synthesized strands[C] to their respective template sequences and tabulated the number of missing, mismatched, and inserted nucleotides (Fig. 2f, Supplementary Fig. 11, Methods). Although multiple alignments exist for several imperfect

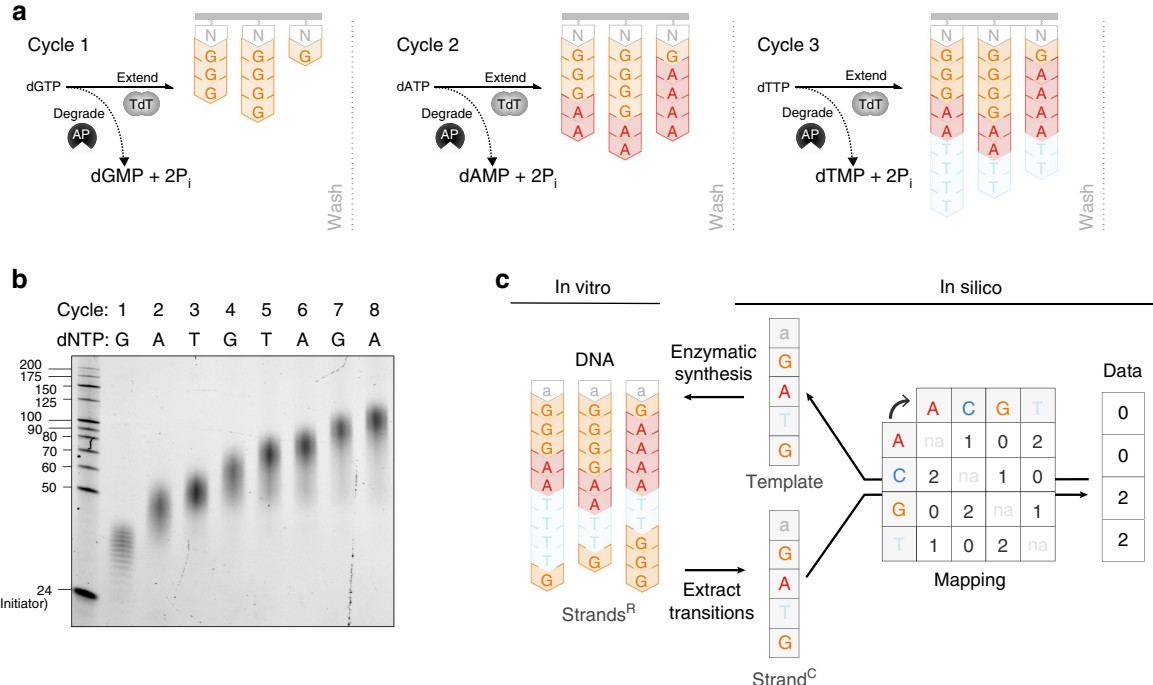

**Fig. 1** An enzymatic synthesis strategy for storing information in DNA. **a** Schematic depiction of a series of enzymatic synthesis reactions consisting of an oligonucleotide initiator (N, gray), terminal deoxynucleotidyl transferase (TdT) and apyrase (AP). The initiator is tethered to a solid support. In each cycle, TdT catalyzes the addition of a given nucleoside triphosphate to the 3′-end of all initiators, whereas apyrase degrades the added substrate to limit net polymerization. A wash can be performed at the end of each cycle to remove reaction byproducts or to facilitate downstream processes. **b** DNA strands synthesized for each of eight consecutive synthesis cycle, as shown on 15% TBE-urea gel. The initiators were not tethered to a solid support and no wash was performed between cycles. The first lane is a single-stranded DNA size marker, which includes 24 nucleotide long initiator oligonucleotide. **c** A schema for interconversion of DNA and information. Raw strands (strands$^R$) represent enzymatically synthesized DNA. A compressed strand (strand$^C$) represents a sequence of transitions between non-identical nucleotides. Transitions between nucleotides, starting with the last nucleotide of the initiator (as an example N = "a", gray) are mapped from the compressed strand to digital data in trits. If a strand$^C$ is equivalent to the template sequence, all desired transitions are present and the information stored in DNA is retrieved

strands$^C$, which ambiguate the exact position of errors, the type of error for each strand$^C$ can be distinguished. Our analysis indicates that 9.5% of strands$^C$ contained one or more mismatches, 10.7% contained one or more insertions, and 66.1% contained one or more missing nucleotides. Thus, the dominant type of error is missing nucleotides in a strand$^C$, which corresponds to a strand that did not get extended by an added nucleoside triphosphate in at least one synthesis cycle.

In spite of synthesis errors, we retrieved information from the pool of synthesized DNA strands$^C$ by applying a simple two-step in silico filter. As each template sequence is designed with a specific architecture (Methods), we first filtered synthesized strands$^C$ by length and presence of a terminal "C". Owing to this filter, the fraction of perfect strands for all template sequences (H01–H12) increased from an average of ~ 19% to an average of ~ 89% (Fig. 2g). We then selected for the most abundantly synthesized strand$^C$ variant in this subset to retrieve data.

We also sequenced H01–H12 strands$^R$ using an entire MinION flowcell (Oxford Nanopore) and observed that the most abundant species, an average of 49.9% of filtered strands$^C$, were perfectly synthesized (Supplementary Fig. 12A). This is largely consistent with results from Illumina sequencing, with the slight decrease likely owing to errors currently inherent to state-of-the-art nanopore sequencing[29]. With these experimental results, we performed simulations to determine what fraction of sequencing resources would have been adequate for robust data retrieval from each of the 12 template sequences H01–H12. We simulated repeated trials which, at a given fraction of the total sequencing

run, randomized the translocation time of each DNA strand$^R$ through the nanopore and assessed whether data could be retrieved (Methods). These simulations indicate that only half of the total sequencing resources were needed to robustly retrieve data from DNA using Oxford Nanopore (Fig. 2h, Supplementary Fig. 12B, Supplementary Table 5).

These results reveal the potential advantage of real-time, rather than batch, sequencing for information retrieval. Whereas the Illumina platform sequences all DNA strands in parallel and reports the outcome in batch, the Oxford Nanopore platform offers asynchronous sequencing by translocation of DNA strands through independent nanopores and streams the outcome. As a result, only nanopore sequencing can be terminated as soon as data are retrieved and remaining reagents provisioned for later use. Further improvements to nanopore sequencing, such as increased DNA translocation speeds[30–33] or selective sequencing[34], may be advantageous for DNA information storage applications.

**Coded strand architecture.** Experimental results indicate that data can be retrieved by in silico filtering to extract perfectly synthesized DNA strands. However, this system suffers from a non-negligible rate of missing nucleotides, which necessitates the synthesis and readout of a large number of strands per template sequence. A large number of strands provides a high level of physical redundancy. To reduce the number of strands stored, we developed a codec to efficiently retrieve information from

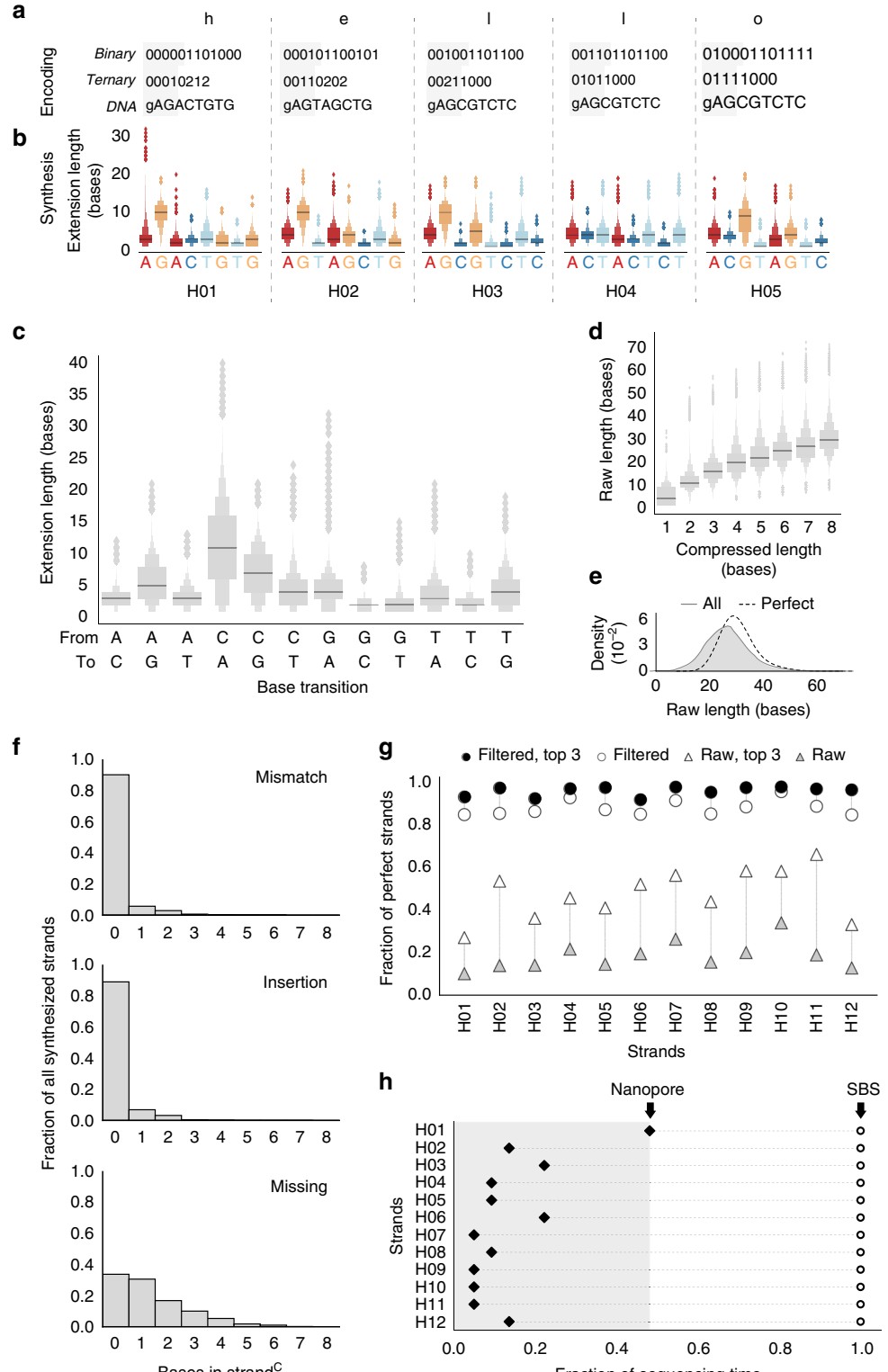

populations of imperfectly synthesized, diverse strands. We utilize statistical inference methods to reconstruct template sequences from diverse strands, each carrying partial information. The codec models information storage in DNA as a communications channel in order to resolve errors accumulated from synthesis, storage, and sequencing (Fig. 3a).

A key feature of this codec is the addition of synchronization nucleotides which are interspersed between information-encoding nucleotides (Fig. 3b). These nucleotides provide redundancy for error correction that is similar to bit-level logical redundancy. However, the redundancy is added to nucleotide sequences instead of bit sequences. Synchronization nucleotides act as a scaffold to aid the reconstruction of a template sequence from DNA strands[C] that may contain missing, mismatched, and inserted nucleotides. As an example, consider a template sequence of eight nucleotides (CTCGTGCT) and two synthesized

**Fig. 2** Demonstration of information storage in DNA using enzymatic synthesis. **a** The message "hello world!" was encoded in 12 template sequences, H01–H12, each representing one character. Transitions between nucleotides start with the last base of the initiator, which is labeled 'g'. A header index (shaded gray) denotes strand order. Only results from H01–H05 are shown (see Supplementary Fig. 9). To encode each character, its respective ASCII decimal value, prefixed with an address is represented in base 2 (binary) or in base 3 (ternary) (see Supplementary Table 2), mapped to transitions (see Fig. 1c), resulting in template sequences with nucleotides to be synthesized (capitalized). **b** Extension lengths for each base from **a** is shown as a letter-value plot with median. Only perfect strands$^R$, those whose strand$^C$ is equivalent to a template sequence, are presented. Synthesis was performed with initiators tethered to beads and sequencing performed on the Illumina platform. **c** Distribution of extension lengths for each nucleotide transition, combined across all positions from all perfect strands is shown as a letter-value plot with median. **d** Stepwise increases in strand$^R$ length with an increasing strand$^C$ length for all synthesized strands of H01–H12 is shown as a letter-value plot with median. **e** Distribution of all strand$^R$ lengths. Distributions are derived via kernel density estimation for all synthesized strands ('all', gray shading) and a subpopulation of strands that contain all desired transitions ('perfect', dotted line). **f** Bulk error analysis for all synthesized strands of H01–H12. All strands$^C$ were aligned, by Needleman–Wunsch, to their respective template sequences, and the number of mismatches, insertions, and missing nucleotides were tabulated. **g** Information retrieval with in silico filtering. Fraction of perfect strands$^C$ is shown before (triangles) or after filtering (circles). Fraction of perfect strands$^C$ is shown for all sequences (white) or only the top three most-abundant sequences (black). **h** Information retrieval by different sequencing platforms. Streaming nanopore sequencing (Oxford, filled diamonds) was compared with batch sequencing-by-synthesis (Illumina, open circles). Each dot indicates the fraction of sequencing run at which each strand is robustly retrieved (100% correct with 99.99% probability). Arrows denote the fraction of the sequencing run at which all data are robustly retrieved using each platform. Source data for **b**–**h** are provided in the Source Data file package

DNA strands$^C$ (CTCTGC and TCGTCT), each with two missing nucleotides. Without a scaffold, data cannot be retrieved as three equally valid reconstructions are possible. By contrast, a scaffold constrains the number of possible sequences to one, allowing data retrieval from otherwise unusable DNA strands$^C$. Although the inclusion of synchronization nucleotides reduces the number of nucleotides allocated for data per template sequence, this design provides error correction and the ability to harness the physical redundancy of diverse DNA strands for data retrieval (Supplementary Note 2).

To reconstruct missing nucleotides from strands$^C$ by scaffolding, the population of synthesized DNA strands for a desired sequence must be sufficiently diverse. Scaffolding can resolve nucleotide errors that occur in different locations, whereas systematic errors require additional forms of error correction. To analyze the diversity of imperfect strands generated from the enzymatic process, we synthesized a longer 16-nucleotide template sequence (called E0) containing 12 unique transitions to mitigate ambiguous alignments (Fig. 3c). We performed in silico size selection of strands$^R$ ranging 32–48 bases in length, assuming that each of the 16 nucleotides in the template sequence was synthesized with an average extension length of two to three bases (Supplementary Fig. 13A). We analyzed this purified set by aligning the corresponding strands$^C$ to the E0 template and observed that missing nucleotides occurring in different locations were the predominant form of error (Fig. 3c, d, Supplementary Fig. 13B), a result consistent with our previous analyses (Fig. 2f). We observed that the median strand$^C$ length was 12 nucleotides and the maximal number of variants occurred at this length. We also calculated the Levenshtein edit distance[35], which summarizes the number of single-nucleotide edits required to repair a strand$^C$. The median edit distance for these variants was four, suggesting that synchronization nucleotides could be placed approximately every three or four nucleotides to recall missing strand$^C$ nucleotides from diversely synthesized strands (Supplementary Note 3, Fig. 3d, Supplementary Fig. 13C). These data provide guidance for devising methods of statistical inference.

Having assessed the scaffolding parameters, we set out to establish a mathematical model for statistical inference that would enable the reconstruction of a template sequence from a population of diverse but imperfect strands$^C$. We adapted a statistical framework known as maximum a posteriori (MAP) estimation[36]. To utilize this framework, we built a Markov model to describe the synthesis of a strand$^C$ with error probabilities for mismatches, insertions, and missing nucleotides, derived from

analyses of the purified set of E0 strands$^C$ (Supplementary Fig. 16A). These state probabilities can be used to score all possible reconstruction solutions consistent with a scaffold, considering mismatches and insertions in addition to missing nucleotides (Supplementary Fig. 17, Supplementary Note 2). Our calculations provide a probability of occurrence for each nucleotide at each position and allow for the generation of a consensus, indicating the most probable nucleotide per position (Supplementary Note 2).

In summary, the digital codec relies on three elements: (i) a coded strand architecture that includes synchronization nucleotides to localize and reduce errors with alignment, (ii) sufficiently diverse strands$^C$ produced by synthesis, and (iii) sequence reconstruction from strands$^C$ based on a mathematical model of DNA synthesis suitable for statistical inference. Given a population of imperfectly synthesized strands, the digital codec reconstructs template sequences via MAP estimation, using synchronization nucleotides as a scaffold.

To test this codec, we encoded and synthesized the message "Eureka!" as four template sequences, E1–E4 (Fig. 4a, Supplementary Note 2). Each template sequence contains a 2-bit address to delineate its order, and 14 bits of data. These 16 bits are encoded in a template sequence of 16 nucleotides, which includes four synchronization nucleotides, resulting in an efficiency rate of 1 bit stored per nucleotide (Supplementary Note 2, Supplementary Fig. 15B). Sequences E1–E4 carry a total of 64 bits of information, including addressing, and were synthesized in parallel on beads with a wash every cycle. Following the last synthesis cycle, strands were ligated to a universal adapter, PCR amplified, and stored as a single pool (Methods).

To reconstruct data from synthesized strands, we first applied in silico size selection of all strands$^R$ of length 32–48 nucleotides (Supplementary Fig. 18). This set of 4521 purified strands$^R$ contained only 31 perfect strands (Supplementary Fig. 20B). We used this purified set with MAP estimation (Supplementary Note 2) and successfully reconstructed template sequences (Fig. 4b). We further determined the minimal number of strands required for template reconstruction and found that only 10 strand$^C$ variants were required, each with an error tolerance of ~30% resulting from missing an average of four or five nucleotides out of 16 total (Supplementary Fig. 26). We also assessed the number of sequencing reads required for a 90% probability of data retrieval and found that all four template sequences were robustly reconstructed with 200, 150, 500, and 100 reads for E1–E4, respectively, with a median of 175 reads (Fig. 4b). Sequence E3 required the most sequencing reads for

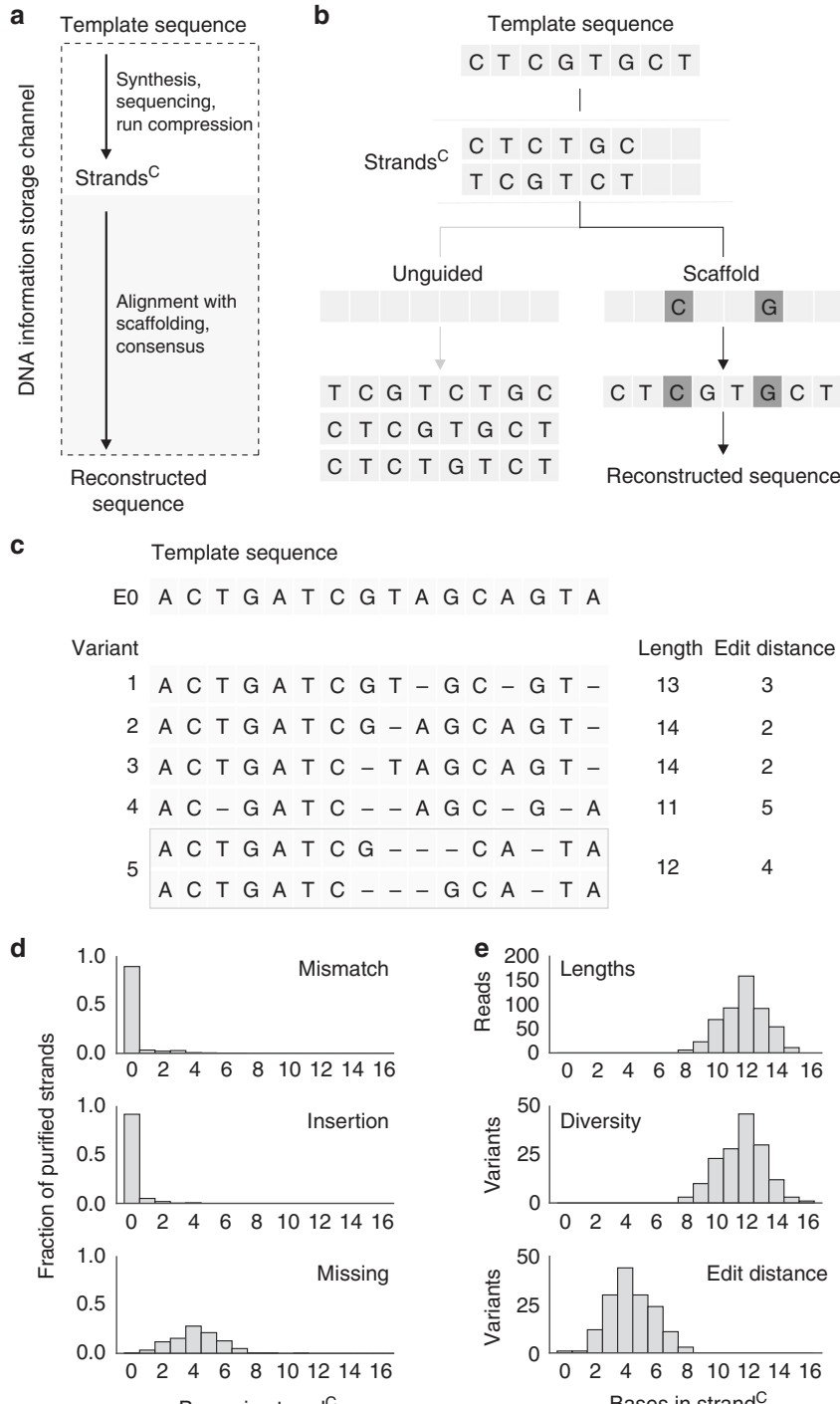

**Fig. 3** Coded strand architecture for sequence reconstruction. **a** A DNA information storage channel. Data are converted to template sequences, synthesized (yielding strands[R]), and can be stored in vitro. Retrieval starts with sequencing, then transitions of non-identical nucleotides are extracted in silico to form strands[C]. Data retrieval occurs when the template sequence and reconstructed sequence are equivalent. Errors that occur in the synthesis and sequencing steps can be modeled as a communications channel. **b** A coded strand architecture, "scaffold", enables data retrieval from strands[C] that are missing nucleotides, whereas an "unguided" reconstruction results in multiple possible solutions. Synchronization nucleotides (dark gray boxes) localize errors to yield a single reconstructed sequence. **c** A 16-base transition sequence, E0, is synthesized and sequenced with Illumina. Examples of diverse strands[C] produced by synthesis of E0. Strands[C] are aligned, by Needleman–Wunsch, to the template. Ambiguous alignments can exist depending on the location and number of missing nucleotides within a strand[C]. **d** Error analysis for purified strands of E0. Synthesized strands were purified in silico, by filtering for strands[R] between 32 and 48 bases in length, and corresponding strands[C] were aligned by Needleman–Wunsch to the E0 template. For each alignment, the number of mismatches, insertions, and missing nucleotides were tabulated. **e** Evaluating the diversity of synthesized strands. The number of sequencing reads for each length of strand[C] was tabulated. Diversity was evaluated as the number of unique variants at each length of strand[C] and the Levenshtein edit distance was computed with respect to the E0 template. The set of 802 purified strands contains two perfect strands. Source data for **c** and **d** are provided in the Source Data file package

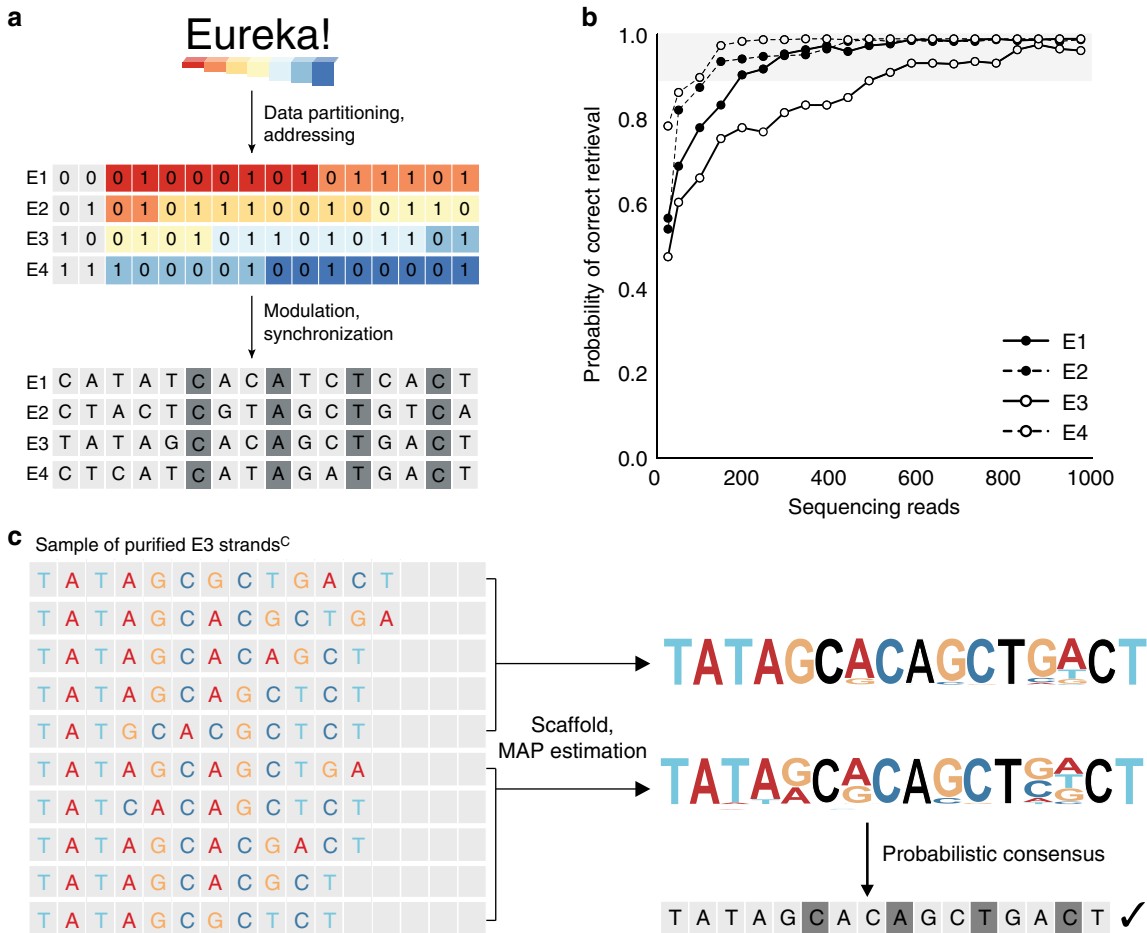

**Fig. 4** Coded strand architecture for robust information storage. **a** The message "Eureka!" was encoded and partitioned into four template sequences, E1–E4. Each sequence stores a 2-bit address and 14 bits of data. These bits are mapped to a template sequence of 16 nucleotides, which includes four synchronization nucleotides (dark gray). Synthesis was performed with initiators tethered to beads and sequencing performed on the Illumina platform. **b** Retrieving information from E1 to E4. Synthesized strands[R] were sequenced using the Illumina sequencing-by-synthesis (SBS) platform and purified in silico based on raw length of 32–48 nucleotides (Methods). The decoding accuracy for each sequence is defined as the probability of 100% correct data retrieval for a given number of reads, estimated over 500 decoding trials. Each trial is based on a randomly drawn set of purified strand[C] variants. A 90% decoding accuracy (gray band) is considered sufficient for robust data retrieval, and this accuracy could be further reinforced by other codec modules. **c** Decoding of E3. A set of 10 DNA strands[C] is decoded as two sets of five strands[C]. The decoder uses MAP estimation and a scaffold to determine the probability for each of the four nucleotides at every position. The decoded sequence is a probabilistic consensus of the reconstructed sequences from MAP estimation and successfully retrieves the data stored in E3. Source data for **b** is provided in the Source Data file package

reconstruction as synthesized strands contained one extra edit on average in comparison with synthesized strands for other template sequences (Supplementary Figs. 19, 20). We found that MAP estimation was a more robust decoding algorithm than our previous two-step filter for H01–H12, requiring fewer reads for data retrieval (Supplementary Fig. 26). The "Eureka!" synthesis experiment shows that a digital codec can support data storage in DNA strands synthesized with error rates that exceed those of current synthesis methods.

**Scalable codec for digital information storage**. If a sufficient number of template DNA sequences are synthesized, byte- and kilobyte-scale storage systems are possible (Fig. 5a). Specifically, the "hello world!" experiment encodes 12 bits per template sequence of eight nucleotides, achieving an efficiency rate of storage of 1.5 bits per template nucleotide. These 12 bits can be allocated for addresses or data. A storage system with a maximum of 256-bytes is possible if 11 bits are used for addressing 2048 template sequences, each storing 1 bit of data. The "Eureka!"

experiment succeeds in storing 16 bits per template sequence of 16 nucleotides, achieving an efficiency rate of storage of 1 bit per template nucleotide. A maximum storage system of 4-kilobytes is possible if 15 bits are used for addressing 32,768 template sequences, each storing 1 bit of data (Supplementary Table 7, Supplementary Note 2).

Through mathematical modeling and simulations, we next assessed the scalability of our digital codec for gigabyte- and petabyte scale storage, under the assumption that the requisite number of DNA template sequences could be synthesized (Supplementary Note 4). Increased storage capacities require additional nucleotides per template sequence to accommodate data, addresses, redundancy for synchronization, and bit-level logical redundancy. For simulations, bit-level logical redundancy was included per template by applying error-correcting codes (ECCs) (Supplementary Note 2). We determined that we could store 36 bits, for addresses and data, in a 74-nucleotide template sequence, and 57 bits in a 152-nucleotide template sequence (Fig. 5a). To arrive at these efficiency rates, randomly generated

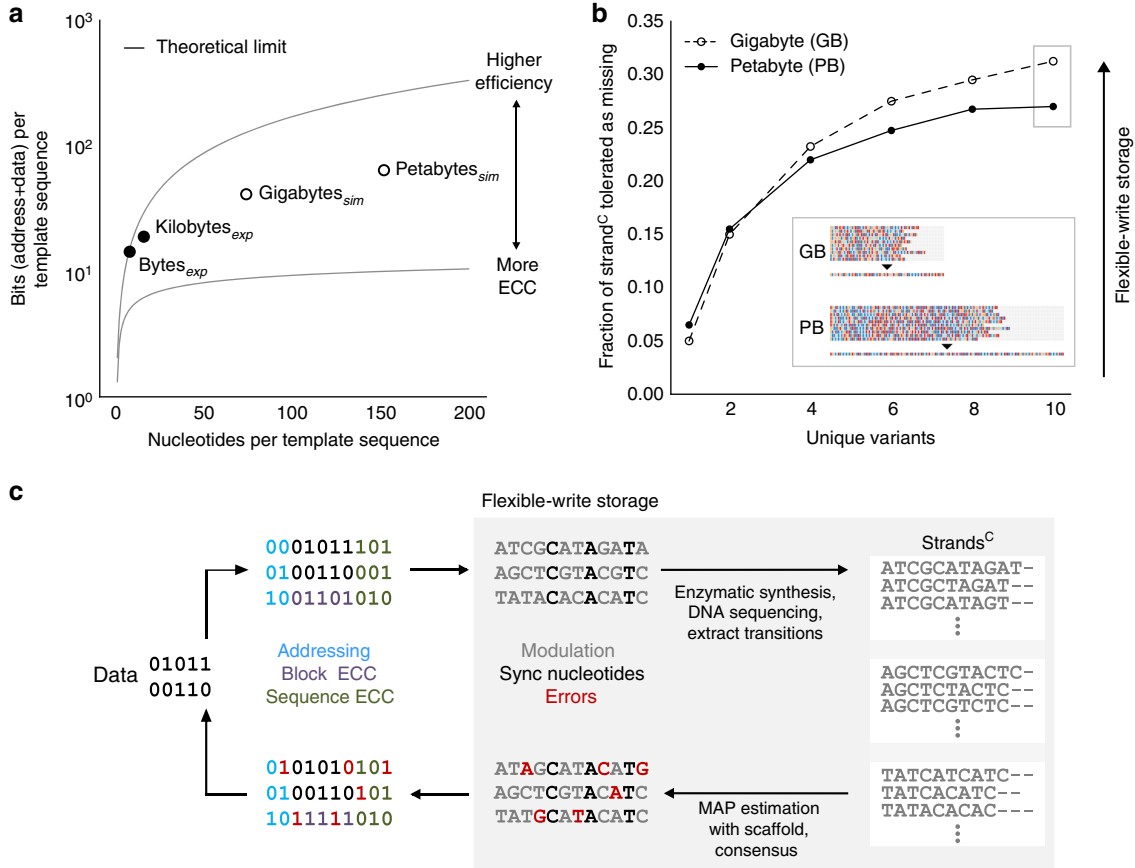

**Fig. 5** A roadmap for scaling DNA storage systems. **a** Efficiency of storage for experimental and simulated systems. Experimental systems (black) include storing 12 bits in 8-nucleotide template sequences, and 16 bits in 16-nucleotide template sequences. Simulated maximum storage systems (white circles) include gigabyte scale that stores 36 bits in a 74-nucleotide template sequence, and petabyte scale that stores 57 bits in a 152-nucleotide template sequence. The amount of bits stored per sequence is dependent on the amount of error-correction codes (ECC) that are applied. Reducing ECCs increases the efficiency rate of storage. The upper bound theoretical limit represents a maximum efficiency of storage of ~1.58 bits per transition between non-identical nucleotides (Supplementary Note 2). The lower bound theoretical limit represents the minimum number of bits per template sequence that must be stored for only addressing (Supplementary Note 4). See all tested storage systems in Supplementary Table 8. **b** Flexible-write storage is enabled by a codec, which harnesses diversely synthesized strands. The decoding pipeline supports robust data retrieval from synthesized strands with a significant percentage of errors. Inset: with ten strand[C] variants, each with ~30% missing nucleotides, the correct decoded sequence can be reconstructed for both gigabyte- and petabyte scale maximum storage capacities. **c** A system architecture for storing information in enzymatically synthesized DNA. A bitstream is partitioned into rows, each augmented with an address to delineate its order for reassembly. An ECC such as a Bose–Chaudhuri–Hocquenghem (BCH) code can be applied to each row, or an ECC such as a Reed–Solomon (RS) code can be applied across multiple rows, to protect data from errors (Supplementary Note 2). Modulation consists of mapping sequences of bits to template sequences, which includes synchronization nucleotides. Enzymatic synthesis then produces multiple diverse strands[C] per template sequence. The resulting strands[C] are used for sequence reconstruction based on MAP estimation and probabilistic consensus. Subsequently, the reconstructed sequence is demodulated into bits. Error-correction is applied to ensure data retrieval. Source data for **b** are provided in the Source Data file package

data were partitioned and mapped to template DNA sequences. Strands[C] were generated in silico using a Markov model for a wide range of synthesis accuracies (Methods). We performed repeated decoding trials with different sets of strands[C] and measured the probability of data retrieval (Methods). Simulations indicate that if at least 10 diverse strand[C] variants are available per template, then ~30% missing nucleotides per strand[C] may be tolerated (Fig. 5b, Supplementary Fig. 28). We found qualitatively similar results when simulated strands[C] also included mismatch and insertion rates exceeding those observed experimentally (Supplementary Fig. 28, Supplementary Note 2). These simulations show that the codec can support a flexible-write approach for distributively storing information in diverse DNA strands at increased storage capacities (Fig. 5c, Supplementary Fig. 30, Supplementary Note 2).

## Discussion

DNA synthesis, sequencing, polymer storage, and amplification can produce errors that will influence the efficiency rate of storage. Our digital codec balances tradeoffs between physical redundancy, synchronization overhead, and bit-level logical redundancy (ECCs). With ideal synthesis accuracies, efficiency rates of storage could reach the theoretical maximum of ~1.58 bits per template nucleotide (Supplementary Fig. 27, Supplementary Note 2). This maximum rate reflects our encoding scheme of storing information in transitions between non-identical nucleotides, compared to an estimated upper bound of ~1.83 bits per template nucleotide[6]. To maintain a high efficiency rate of storage, as seen in the "hello world!" experiment, a high level of physical redundancy and sequencing is required. In the "Eureka!" experiment, the digital codec includes synchronization

overhead in order to reduce sequencing coverage. For large-scale simulated systems, our analyses indicate that efficiency rates must be reduced ~ 6.5-fold and ~ 8.4-fold for gigabyte- and petabyte scale systems, respectively (compared with ~ 1.58 bits per template nucleotide) (Fig. 5a). In exchange, the digital codec resolves several types of errors, including missing nucleotides in synthesized strands$^C$, which would otherwise drastically reduce storage capacities and prohibit data retrieval altogether[37,38] (Supplementary Note 2).

Taken together, these results demonstrate a proof-of-concept strategy for enzymatic synthesis and a digital codec to accurately store information in DNA without requiring single-base precision. However, this approach comes at a cost to volumetric storage density owing to introduced redundancy. Specifically, extension lengths incur approximately threefold loss (Supplementary Fig. 18). Synchronization and logical redundancies incur more than sixfold loss as seen in large-scale simulations (Supplementary Note 2). Furthermore, for our digital codec, we require a physical redundancy of ~ 10 diverse DNA strands (each with < 30% error), which also incurs a corresponding volumetric density loss. The exact minimum amount of sequencing depth required to obtain sufficient physical redundancy for data retrieval remains an open question. To address this, further work that includes stringent physical purification of informative strands and synthesis of a larger number of template sequences will be required. For data retrieval using highly purified DNA synthesized with phosphoramidite chemistry, the required sequencing depth ranges between 4 and 14 reads (Illumina) and > 80 reads (Oxford Nanopore) using alternative decoding methods[7]. Overall, these reductions in volumetric density owing to error-correction overheads may be acceptable, as the theoretical maximum volumetric density of DNA is three to six orders of magnitude denser than the projected limit of baseline memory technologies[2].

Importantly, several advantages to read/write speed and cost may arise from this approach. For writing, our reagent cost analyses indicate that enzymatic synthesis can be a cheaper alternative to the phosphoramidite process with equivalent feature sizes (Supplementary Fig. 35a). Further miniaturization, together with reductions to enzyme cost through recycling, provide a potential roadmap for overall reduction in reagent costs by several orders of magnitude (Supplementary Fig. 35, Supplementary Note 6). In addition, storage capacities may be increased as TdT can add ~ 500 (Supplementary Fig. 4B) to thousands[23] of nucleotides per strand, to enable the use of longer template sequences. Furthermore, synthesis times may be reduced as this kinetic system circumvents the need for blocking moieties typically required for producing single-base precise DNA[11,17] that often result in longer cycle times (Supplementary Table 6, Supplementary Note 6). Recently, other conceptually similar work has been posted[39] which also uses natural nucleoside triphosphates to facilitate DNA synthesis for storage applications. For reading, synthesized strands with short homopolymeric stretches may be advantageous when using specialized readout technologies such as nanopore sequencing, which can translocate DNA at higher speeds if single-base resolution is not required[30–33,40]. More broadly, this flexible-write system exploits the distributive nature of TdT polymerization[41] to store information across multiple strands for a given template sequence. As a result, biosecurity concerns associated with widespread DNA synthesis may be alleviated as genes are unlikely to be produced.

To address the challenge of increasing storage capacities in DNA, industrial-grade automation and developments in biochemistry will be needed. For example, the number of template sequences must increase by orders of magnitude for large-scale storage (Supplementary Note 4). As a significant first step, we demonstrate that this enzymatic synthesis process can be translated from beads in solution to a planar, solid support, microarray format. With a 2D-array prototype, we synthesized three unique template sequences (S01–S03, each with 13 cycles) in triplicates as spots on a glass slide and achieve synthesis accuracies similar to those observed in solution (Supplementary Figs. 31–34, Supplementary Note 5). Further hardware engineering for parallelization and automated fluid handling, as well as optimization of surface chemistries to ensure functional interactions between DNA and proteins will also be required. Improvements to this biochemistry will increase synthesis accuracies. Other nucleotide analogs beyond that presented in this work may be useful for minimizing extension lengths (Supplementary Fig. 8, Supplementary Note 1) or for reducing the formation of DNA secondary structures. It may be necessary to engineer TdT mutants capable of operating in conditions which denature DNA, such as high temperatures, to ensure the accessibility of the free 3′ hydroxyl end for polymerization. To protect against specific errors, the digital codec may be tuned. For example, the modulation block of the codec can reduce the use of the "G" nucleotide to reduce the occurrence of G-quadruplexes. Overall, additional technological achievements will improve upon this enzymatic-based synthesis platform and inform the design of a complete read and write system for digital information storage in DNA.

## Methods

**"hello world!" experiment**. An initiator oligo (5Am12-fSBS3-acgtactgag, Supplementary Table 1) was immobilized on 5.28 micron carboxyl polystyrene beads (Spherotech CP-50–10) using carbodiimide conjugation. 5 mg beads were washed twice in 100 mM 2-($N$-Morpholino)ethanesulfonic acid (MES) buffer pH = 5.2 and resuspended in 100 μl of the same buffer. The oligo, 5Am12-fSBS3-acgtactgag, was resuspended at 100 μM in water. A 1.25 M batch of EDC was prepared by dissolving 120 mg EDC (Sigma E1769, from − 20C storage) in 500 μl of 100 mM MES pH = 5.2. 40 μl of the 1.25 M EDC batch was mixed with 30 μl (3 nmole) of the 5Am12-fSBS3-acgtactgag oligo and 30 μl of 100 mM MES pH = 5.2 and added to the beads and mixed by vortexing for 10 sec. The suspension was rotated at room temperature overnight. After incubation overnight, the beads were washed three times with 1 mL buffer containing 250 mM Tris pH 8 and 0.01% Tween 20, each time rotating at room temperature for 30 min. The beads were then resuspended in 500 μl Tris-ethylenediaminetetraacetic acid buffer with 0.01% Tween 20 and stored at 4 °C until use.

For each character, the ASCII decimal (data) was converted to base 2 (for binary, 8 bits) or to base 3 (for ternary, 5 trits). Similarly, the addresses were converted from a decimal value to base 2 (for binary, 4 bits) or base 3 (for ternary, 3 trits). Addresses were concatenated to data to form a resulting string of 12 bits or 8 trits. A custom Python script was used to map trits to template sequences H01–H12 shown in Supplementary Table 2. Nucleoside triphosphates (Invitrogen) were prepared at the following concentrations: 8 mM dATP, 4 mM dCTP, 4 mM dGTP, and 16 mM dTTP. For each template sequence (Supplementary Table 2), the required dNTP volumes corresponding to each transition type were dispensed (Supplementary Table 4) in a 96-well PCR plate (VWR) using a Mantis liquid handler (Formulatrix), which has a minimum dispense volume of 0.2 μL. Once the dNTPs were loaded, 30 μg of initiator-conjugated polystyrene beads for each of the twelve template sequences were suspended in an enzymatic reaction mix, comprised of 1 × Custom Synthesis Buffer (14 mM Tris-Acetate, 35 mM Potassium Acetate pH 7.9, 7 mM Magnesium Acetate, 0.1% Triton X-100, 10% (w/v) PEG 8000) with 1U/μL TdT (Enzymatics) and 1mU/μL apyrase (NEB). For each synthesis cycle, beads that are suspended in the enzymatic reaction mix are exposed to dNTPs, by transfer to the subsequent well, with a multichannel pipettor. Reactions are incubated at room temperature for one minute. Every two cycles, the beads were collected by centrifugation (3 min at 1310 g), washed with 1 × Custom Synthesis Buffer without PEG, collected by centrifugation, and resuspended in fresh enzymatic mix. Following addition of the last dNTP for each sequence, a poly-deoxycytidylate (polyC) tail was synthesized by addition of 1 μL of 1.6 mM dCTP (32 μM final) to the enzymatic mix to enable efficient ligation. Afterwards, beads were collected by centrifugation then washed with 10 mM Tris-HCl pH8.0 with 0.1% Triton X-100, and resuspended in 10 μL of the same buffer.

A universal adapter was ligated to the 3′ of the synthesized strands using a hybridization-based strategy[42]. The 5P-rSBS9-GGG adapter (Supplementary Table 1) forms a hairpin with a 5′ dGTP tail overhang, which hybridizes to single-stranded DNA strands ending in a polyC tail. The beads with polyC-tailed synthesized DNA were resuspended in a reaction composed of 1 μM 5P-rSBS9-GGG adapter, 1X T4 DNA Ligase Buffer (NEB), 20% PEG 8000 (Sigma), 500 mM Betaine (Sigma), and six units of T4 DNA Ligase (Enzymatics) per μL. The ligation mixture was incubated at 16 °C overnight. After ligation, beads were washed twice

with 100 μL of 10 mM Tris-HCl pH8.0 with 0.1% Triton X-100 and resuspended in 100 μL of 10 mM Tris-HCl pH 8 with 0.01% Tween 20. Then, 5 μL of each strand was amplified with primers tSBS3 and tSBS9 in a 10 μL reaction with cycle-limited real-time PCR for 15 cycles and column purified (Zymo).

For Illumina sequencing, amplified strands were diluted and used as a template for a PCR reaction with NEBNext Dual Indexing Primers. Each strand received a different index by real-time cycle-limited PCR for 15 cycles. Barcoded strands were then combined and sequenced single end using Illumina MiSeq v2 150 Micro. For Oxford Nanopore sequencing, 1 μL of each Illumina-barcoded strand was diluted 100-fold in Tris-HCl pH 8.0 with 0.01% Tween 20 and amplified with nested primers, comprising a barcoding primer pair, LWB 01–12 from SQK-LWB001 (Oxford Nanopore), and 50 nM of primers PR2-P5 and 3580F-P7 (Supplementary Table 1) for 10 cycles. 5 μL of each strand was then pooled (60 μL total) and cleaned with 90 μL of Agencourt Ampure XP beads according to the manufacturer's protocol. One microliter of the pooled library was diluted with 9 μL of 10 mM Tris-HCl pH8 with 50 mM NaCl (10 μL total). One microliter of Rapid 1D Sequencing Adapter (Oxford Nanopore) was added, flicked, and incubated for 5 min at room temperature. A total of 11 μL of this presequencing mix was combined with 30.5 μL of Running Buffer with Fuel Mix (RBF), 26.5 μL of library loading beads, and 7 μL of water, added to a R9.4 flow cell, and run with SQK-LWB001 settings for 48 h of sequencing.

For analyses of synthesized strands and data retrieval from Illumina sequencing, demultiplexed reads were first trimmed with cutadapt 1.9.1[43], with an error tolerance up to 10%, to remove the 5′ initiator oligo sequence and the 3′ universal oligo sequence. Only reads containing both sequences for trimming were retained for further analysis. Custom Perl or Python scripts were used to process these trimmed reads. Sequences of non-identical nucleotides were extracted from each read by run-length compression[44] and the occurrence of each unique sequence was tabulated. To determine the type of synthesis errors, each strand was aligned to its respective sequence using Needleman–Wunsch algorithm[45], implemented as pairwise2 in Biopython 1.70, with match, mismatch, gap-open and gap-extension scoring set as 2, −3, −5, and −5, respectively. For each alignment, the number of mismatches, insertions, and missing nucleotides were tabulated. For data retrieval, a two-step filter was used. The first step is to filter for the designed number of nucleotides, which contain a terminal 'C', used for ligation, in compressed strands. As a result, 8 of 12 template sequences, specifically H01, H02, H04, H08, H09, H10, and H11, have nine nucleotides to be synthesized. In contrast, 4 of the 12 template sequences, specifically H3, H5, H6, and H7, contain only eight nucleotides to be synthesized. The second step is to select the most frequently synthesized compressed strand variant.

For analyses of synthesized strands and data retrieval from Oxford Nanopore sequencing, base calling, and demultiplexing was performed with Albacore 1.2.6 using the configuration file for 1D reads at 450 bp per second using the R9.4 chemistry to match the flowcell FLO-MIN106 and kit SQK-LWB001. Demultiplexing was further verified with Porechop 0.2.2[46] with default settings for quality control. The resulting reads were trimmed with cutadapt as described above, except with an empirically determined increased error tolerance to compensate for the higher error rate observed for nanopore sequencing. Strands can be sequenced in either orientation. Accordingly, we found that an error tolerance of 25% resulted in no > 50% of strands being trimmed. Only reads that presented both sequences for trimming were retained for further analysis. Reads in the opposite orientation were not processed. Data retrieval for each sequence was performed as above for Illumina sequencing with a two-step filter.

Real-time data reconstruction with nanopore sequencing reads was simulated by applying the two-step data retrieval filter to a subsampled number of shuffled sequencing reads obtained up to a given time point. The 48-hour sequencing run was split into 2-hour increments. For each increment, the timestamp for all reads obtained during the entire sequencing run were shuffled and the number of reads corresponding to the total elapsed sequencing time up to the given increment were randomly sampled. We assessed the probability of correct retrieval by performing 10,000 decoding trials for each increment and expressed each time interval to fraction of total sequencing time.

**"Eureka!" experiment**. Methods of encoding and decoding, including sequence reconstruction from synthesized strands, were implemented according to specified and listed mathematical equations described in the Supplementary Notes. The decoding pipeline for the experiment included MAP estimation and consensus algorithms for sequence reconstruction, and was implemented in the C++ programming language. The computer code was compiled via a g++ compiler on an Ubuntu Linux operating system.

The message "Eureka!" consisting of seven ASCII characters, equivalent to 56 bits of payload data, was encoded as four template sequences (E1–E4), each containing 16 nucleotides. The encoding steps consisted of data partitioning, addressing, and modulation of bit sequences to nucleotide sequences with no repeated bases (i.e., self-transitions). Modulation included the placement of synchronization nucleotides within DNA sequences as described in the Supplementary Note 2. In addition to E1–E4, sequence E0 was designed for the purpose of error analyses. After enzymatic synthesis of E0 and E1–E4, run-length compressed DNA strands were provided as input to MAP estimation and consensus algorithms for template sequence reconstruction. Reconstructed E1–E4 DNA sequences were demodulated into bit sequences, and payload data were extracted by ordering according to addresses.

The initiator oligonucleotide Bio-U-LT2 (Supplementary Table 1) was conjugated to streptavidin beads (Invitrogen) according to manufacturer instructions at 20% binding capacity and Biotin-14-dCTP was used to bind remaining free streptavidin. Blank beads, which have free streptavidin bound by Biotin-14-dCTP were also prepared. Prior to use, the initiator-conjugated beads were diluted 10-fold with blank beads and washed with 1 × Custom Synthesis Buffer without PEG.

Synthesis of E0–E4 was performed similarly as described above. However, Bromo-dCTP was used instead of dCTP (Supplementary Note 1, Supplementary Fig. 8) and concentrations of each dNTP regardless of transition type were fixed. The final concentration of dNTPs for each cycle were as follows: 10 μM dATP, 15 μM Bromo-dCTP, 5 μM dGTP, and 15 μM dTTP. As above, a series of dNTPs were dispensed for each nucleotide of the template sequence in a 96-well PCR plate. Once the dNTPs were loaded, 100 μg of initiator-conjugated magnetic beads were suspended in the enzymatic reaction mix, comprised of 1 × Custom Synthesis Buffer (14 mM Tris-Acetate, 35 mM Potassium Acetate pH 7.9, 7 mM Magnesium Acetate, 0.1% Triton X-100, 10% (w/v) PEG 8000) with 1U/μL TdT (Enzymatics), and 0.25mU/μL apyrase (NEB). For each synthesis cycle, beads that were suspended in the enzymatic reaction mix were exposed to dNTPs, by transfer to the subsequent well. At every cycle, each reaction was pulse vortexed and incubated for 30 sec at room temperature. Beads were collected by magnet and washed in 1 x Custom Synthesis Buffer without PEG and resuspended with fresh enzymatic mix. The reaction mixture was then transferred to the next well containing the next nucleotide substrate. Following the last cycle, each sample was prepared for Illumina sequencing as described above. Complete Illumina sequencing adapters were added by real-time cycle-limited PCR for 12 cycles. Barcoded strands were then combined and sequenced as single end 175 bp reads using Illumina MiSeq v2 Nano. Sequences were trimmed as before to remove the 5′ initiator oligo sequence (Bio-U-LT2) and the 3′ universal oligo sequence (5P-rSBS9-GGG, Supplementary Table 1). Only reads that presented both sequences for trimming were retained for further analysis. A sequence of non-identical nucleotides for each raw strand was extracted as above. Purified strands were obtained by selecting strands with raw lengths between 32–48 bases, corresponding to average extension lengths of two to three per template nucleotide. Purified strands were used for analysis of synthesis errors with Needleman–Wunsch and for sequence reconstruction of E1–E4 with the decoding pipeline.

Following purification, DNA strands synthesized for each template sequence E1–E4 were randomly sampled from data according to a target number of reads, and then subject to a two-step filter. A filter was first applied to include those DNA strands with read counts either 1, 2, 3, 4, or 5 depending on the target number of reads, to exclude aberrant DNA strands, which could arise from combinations of synthesis and sequencing errors. A second filter was applied to rank DNA strands according to compressed strand lengths. A total of 10 top-ranked DNA strands were selected from all purified and filtered strands. These 10 strands were used to reconstruct each template sequence using MAP estimation and consensus implemented according to equations in Supplementary Note 2. We assessed the probability of correct retrieval of each template sequence E1–E4 by performing 500 decoding trials for each target number of reads. Each trial consisted of a random sampling of purified reads.

**Simulated large-scale storage systems**. To simulate DNA storage systems, $\Omega$ bits of data and addresses per template sequence were randomly generated. A Bose–Chaudhuri–Hocquenghem (BCH) code was applied for error correction to convert $\Omega$ random bits to $B$ bits per template sequence. The BCH code was computed via MATLAB's (MathWorks) built-in standard BCH encoder. The $B$ bits were mapped and modulated into $K$ nucleotides per template DNA sequence according to designed modulation schemes as explained in the Supplementary Notes. The following DNA storage systems of increasing storage capacity were simulated, represented by codec parameters: $(K, B, \Omega) = (38, 33, 23)$, $(K, B, \Omega) = (74, 63, 36)$, and $(K, B, \Omega) = (152, 128, 57)$.

In order to simulate enzymatic synthesis, a Markov model was used to produce compressed strands. For these, $K$ nucleotides per template DNA sequence were subject to errors including missing nucleotides (deletions), insertions, or substitutions. Synthesis accuracy was varied by simulating a range of error probabilities, primarily for missing nucleotides in compressed strands. Each type of error was assumed to occur independently per nucleotide within a strand. Each strand was produced independently according to the same error statistics.

The codec was tested by performing 500 decoding trials for varying levels of synthesis accuracies. For each decoding trial, a template sequence was randomly generated, and ten compressed strands were synthesized by simulation with the Markov model. These compressed strands were used towards reconstruction of the template sequence via MAP estimation and probabilistic consensus. Each reconstructed sequence of $K$ nucleotides was demodulated into $B$ bits and decoded with a MATLAB (MathWorks) BCH decoder to yield $\Omega$ bits. The probability of correct data retrieval for a specific level of synthesis accuracy was computed as the fraction of successful decoding trials. Results for data retrieval were benchmarked on a multi-core server.

**Reporting summary**. Further information on research design is available in the Nature Research Reporting Summary linked to this article.

## Data availability

Raw sequencing data for synthesis experiments are available from NCBI SRA under SRP185459: H01–H12 (SRR8556774, SRR8556773, SRR8556778, SRR8556777); S01–S03 (SRR8556776, SRR8556775); E0–E4 (SRR8556779). Source data file include processed sequencing data for Figs. 2, 3c–e, 4b–c, 5b and Supplementary Figs. 9–13, 18–29, and 32–34. All other data are available from the authors upon reasonable request.

## Code availability

Processed data and custom code for error analyses, encoding, and decoding are available via https://github.com/citizenlee/DNAinfostorage. Code for the digital codec is available from the authors upon request.

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

## Acknowledgements

We acknowledge John Aach, Nili Ostrov, and Javier Fernández Juárez for helpful discussions and comments on the manuscript, Calixto Saenz and the HMS Microfluidics/Microfabrication Facility, HMS Biopolymers Facility, and Dmitry Rodionov of Formulatrix, Inc. for technical support. This work was supported by funding from National Institutes of Health Grant R01-MH103910-02 (to GMC), Department of Energy Grant DE-FG02-02ER63445 (to G.M.C.), and AWS Cloud Credits for Research program (to H.H.L., G.M.C,).

## Author contributions

H.H.L., R.K., N.G., J.B. and G.M.C. conceived the study. H.H.L., R.K. and G.M.C. developed enzymatic synthesis. N.G. developed the codec. H.H.L., R.K. and N.G. performed experiments and analyzed the data. H.H.L., N.G. and R.K. wrote the manuscript. All authors edited and reviewed the manuscript. G.M.C. supervised the study.

## Additional information

**Competing interests:** H.H.L., R.K., and G.M.C. have filed patents covering the synthesis process (WO 2017/176541) and the encoding/decoding process (PCT/US18/56900). N.G and J.B. have filed a patent for the use of synchronization markers for the codec (WO 2018/148260).

