## [Peer Review File · Nature Communications]

Reviewers' Comments:

Reviewer #1:

Remarks to the Author:

This paper proposes a DNA data storage approach more appropriate for the current state of enzymatic synthesis and nanopore sequencing by encoding data in the transition between bases. This way, "runs" of repeated nucleotides have a lower effect on the ability to retrieve information.

First, the upshot of my opinion is that, based primarily on the basic idea, I believe there is a solid contribution to the field and suggest that the paper be accepted. However, it is just too bad that the amount of data is just really, really, small. The paper would benefit a lot from a candid discussion of the challenges why it isn't larger. I am assuming the authors tried it.... So, why didn't it succeed? I would still advocate to accept it even if there are significant but not insurmountable challenges. I am not suggesting that it should be close to the phosphoramidite-chemistry-based demonstration. Just an order of magnitude or so more to show that the practical challenges aren't fundamental showstoppers. If each template sequence could be demonstrated to hold about 10-20 bytes, that would be comparable to prior studies on a per-strand bases and would make the work be a lot more jaw-dropping.

Now on more detailed comments:

- * Can you show just raw bit error rates with and with the "synchronizaton" markers?
- * Why didn't you include an inner code for logical redundancy? the proposed error correct as described seems to rely entirely on reconstruction from multiple observations which is pretty expensive because it uses more sequencing bandwidth and more physical copies of sequences.
- * The density discussion around line 266 is confusing. It seems that "redundancy" in line 271 refers to physical redundancy. And the need for higher physical redundancy is tied to any logical redundancy, which directly affects coding rate. I suggest clearly articulating coding rate, physical redundancy needs, etc.

Reviewer #2:

Remarks to the Author:

Lee et al. demonstrate a proof of concept use of a de novo enzymatic synthesis strategy that builds on the work of Palluk et al. 2018 overcomes quality and quantity limitations of the widely used phosphoramidite chemistry. They use the TdT polymerase to catalyze linkage of dNTPs with homopolymeric extensions, controlling polymerization with apyrase. Due to the repetitive nature of the homopolymer units, information is encoded in transitions between non-identical nucleotides. Synthesis costs are the primary bottleneck in making DNA data storage scalable and affordable. This approach has many advantages over phosphoramidite chemistry, particularly lower cost, shorter synthesis time, and longer template lengths.

Error-prone oligo synthesis is costly due in large part to the need to clone and sequence verify. The major shortcomings of TdT for industrial scale oligo synthesis can be tolerated in molecular data storage. To address the high error rate (primarily deletions), Lee et al. developed an error correction codec to reconstruct template sequences from incompletely synthesized strands by a scaffolding approach that uses synchronization nucleotides to locate errors. Lee et al. were able to robustly retrieve data from imperfect synthesis using a maximum a posteriori estimation of each nucleotide.

The authors demonstrated the use of nanopore sequencing over SBS. The latter is more amenable to widely used short oligo synthesis approaches, but increased template lengths will benefit from nanopore sequencing and reduction in latency, particularly without the need for single base resolution. Reaching the massive parallelization required for large-scale storage in DNA will require further improvements but Lee et al. discuss the scalability of enzymatic synthesis and demonstrate

a prototype of an array-based platform.

The main bottleneck to large-scale molecular storage is the increasing number of nucleotides needed for indexing the data. The scalability of the approach to Gb/Pb is estimated from simulations, demonstrating the tradeoff between efficiency and increased error correction. Given the unmatched density of DNA data storage, this technology is will likely be applied to very large-scale data, which will require massive parallelization of synthesis, well beyond this proof-of-concept.

Lee et al. demonstrate considerable effort at optimizing the kinetic control of homopolymeric extension. However, a major disadvantage of phosphoramidite synthesis is the need to assemble templates > 200-300 nt. The authors cite the advantage of enzymatic synthesis to yield longer products but they do not demonstrate this important application beyond size separation of extension products. The formation of secondary structures, particularly for G homopolymers, which can contribute to the high deletion rate of TdT synthesis and loss in volumetric density warrant further consideration beyond the small scale analysis for application of in a scalable storage workflow.

The authors demonstrated a tolerance for high error rate but scaling up would require increased sequencing coverage, as demonstrated by the number of reads necessary for robust data retrieval of the 'Eureka!' encoding. Further, both homopolymer runs (>4nt) and high/low GC content show high dropout rates in PCR/sequencing.

Minor points

The supplement is extremely long and there is a lot of cross reference between the main text and supplement.

Given the importance in synthesis cost reduction for scalable DNA storage, I find it strange that this is hardly discussed in the main text.

A nucleotide or NTP is a nucleoside triphosphate, not nucleotide triphosphate

Point-by-point Response

Please view all original, unmodified comments by reviewers in black. Similarly, please view our responses to all questions highlighted in blue.

Reviewer #1 (Remarks to the Author):

This paper proposes ad DNA data storage approach more appropriate for the current state of enzymatic synthesis and nanopore sequencing by encoding data in the transition between bases. This way, "runs" of repeated nucleotides have a lower effect on the ability to retrieve information.

First, the upshot of my opinion is that, based primarily on the basic idea, I believe there is a solid contribution to the field and suggest that the paper be accepted. However, it is just too bad that the amount of data is just really, really, small. The paper would benefit a lot from a candid discussion of the challenges why it isn't larger. I am assuming the authors tried it.... So, why didn't it succeed? I would still advocate to accept it even if there are significant but not insurmountable challenges. I am not suggesting that it should be close to the phosphoramidite-chemistry-based demonstration. Just an order of magnitude or so more to show that the practical challenges aren't fundamental showstoppers. If each template sequence could be demonstrated to hold about 10-20 bytes, that would be comparable to prior studies on a per-strand bases and would make the work be a lot more jaw-dropping.

We agree with the reviewer that synthesizing longer lengths and a greater number of DNA strands would make the work a lot more jaw-dropping. To the best of our knowledge, the only other recently published study of enzymatic synthesis, which presents an alternative method, reports two strands: one of 10 cycles and the other with 3 cycles (Palluk et al. 2018).

For this study of enzymatic synthesis, we have contributed a significant amount of data on TdT biochemistry (Figs. S1-S13, S18-S25, S31-S34). To demonstrate enzymatic synthesis, we report 8 cycles of synthesis for each of 12 DNA sequences (H01-H12) as well as 16 cycles of synthesis for each of 5 DNA sequences (E0-E4). Furthermore, we have demonstrated 13 cycles of synthesis for each of 3 template DNA sequences in triplicates (S01-S03) with a 2D array prototype. The synthesized sequences differ in identity. Additionally, we have also shown that TdT can add at least ~500 nucleotides per DNA strand in our reaction conditions (Fig. S4B). Taken together, these significant results and data are useful for further development of the biochemistry and automation.

More synthesis has not been carried out simply because automated solutions must be developed in order to synthesize a large number of DNA strands in parallel. Over the past few decades, such facilities have been designed and implemented for synthesis using phosphoramidite chemistry. Due to the accessibility of these facilities, nearly all published studies demonstrating information storage in DNA have benefitted by outsourcing the large-scale production of DNA strands to commercial entities such as Agilent or Twist (Church, Gao, and Kosuri 2012; Goldman et al. 2013; Blawat et al. 2016; Erlich and Zielinski 2017;

Organick et al. 2018). By contrast, automation tailored for parallelization of enzymatic synthesis does not exist. Towards the goal of automation and parallelization, we have prototyped a 2D array-based platform (Figs. S31-S34) which represents a significant achievement in translation from a bead-based approach. Further automation is still required for an experimental demonstration of large-scale enzymatic synthesis of DNA. While this demonstration is outside the scope of the current manuscript, it is the goal of current and future work.

Now on more detailed comments:

- * Can you show just raw bit error rates with and with the "synchronizaton" markers?
- * Why didn't you include an inner code for logical redundancy? the proposed error correct as described seems to rely entirely on reconstruction from multiple observations which is pretty expensive because it uses more sequencing bandwidth and more physical copies of sequences.

Bit error rates imply lossy reconstruction of data, and depend on coding schemes such as modulation and demodulation selected as part of the digital codec. We consider only lossless reconstruction of data for which the bit error rate vanishes to zero by design. Our analysis focuses on nucleotide error rates and the probability of lossless reconstruction of data. The digital codec ensures that the probability of lossless reconstruction is close to 1, in exchange for added redundancy (Fig. S30, Supplementary Text 2).

Data storage in DNA is uniquely defined by nucleotide error rates (deletions, substitutions, and insertions of nucleotides), and the amount of sequencing coverage per template sequence (number of diverse DNA strands sequenced per template sequence). An optimal digital codec maximizes the efficiency rate of storage, which is the number of bits stored per nucleotide on average, while ensuring that the probability of lossless reconstruction is close to 1.

Bit error rates for a multi-stage digital codec depend on which stage, after the demodulation of nucleotides to bits, is being measured (Fig. S30). In contrast to bit error rates in synchronized wireless communication systems, bit error rates in DNA storage systems include deletions, substitutions, and insertions of bits. Consider a hypothetical storage system, and modulation scheme: 00 = A, 01 = T, 10 = C, 11 = G. A data sequence 00011011 is mapped to a DNA template sequence ATCG. After DNA synthesis, if a DNA strand is sequenced as ATG and demodulated, the result is a shortened bit sequence 000111. In this case, it is likely that 2 bits have been deleted. However, this bit error rate is a function of the modulation and demodulation scheme, as well as other coding schemes that may be applied. To avoid such ambiguities, we focus on the end-to-end probability of lossless data retrieval.

We have provided extensive analysis of nucleotide error rates for enzymatic DNA synthesis (Figs. 2F, 3C-3E, S11, S13, S19-S25). As illustrated by a Markov model (Fig. S16), even nucleotide error rates depend on a particular choice of alignment algorithm for the evaluation of nucleotide errors. We applied Needleman-Wunsch alignment of DNA strands to the ground truth template sequence to measure nucleotide error rates. Nucleotide error rates are independent of the digital codec and synchronization.

We now address the part of the reviewer's question regarding the effect of synchronization markers on the ability to retrieve data. We note that for the "hello world" experiment (H01-H12 templates),

synchronization was not applied, and filtering of DNA strands was sufficient to recover data losslessly. For the “Eureka!” experiment (E0-E5 templates), both filtering and synchronization were applied to recover data losslessly (Fig. 4B). For the “Eureka!” experiment, the probability of lossless data retrieval improves for a given number of reads when applying synchronization by MAP decoding compared to only filtering (Fig. S26). By design, the “Eureka!” experiment highlights the advantages of harnessing the physical redundancy of DNA produced by TdT synthesis. In simulations of scalable DNA storage systems, physical redundancy is critical for lossless data retrieval, especially if a significant number of deletions occur per strand.

An inner code for logical redundancy was not applied in either the “hello world” experiment (H01-H12 templates) or the “Eureka!” experiment (E0-E5 templates). The primary reason is that an inner code requires a sufficiently long block length to be effective for error-correction, otherwise it is inefficient in terms of the amount of redundancy it requires. Thus, we included an inner code only for simulated DNA storage systems which involve longer sequences.

We analyzed the effect of no synchronization vs. synchronization on recovering data losslessly from either 1, 2, 4, 6, 8, or 10 diverse DNA strands in simulations (Fig. S28). In these simulations, it is shown that data can be reconstructed losslessly per template sequence even from 1 single DNA strand (i.e., no synchronization between strands). This is possible if an inner code for logical redundancy is utilized for error-correction. An inner code can correct several errors, but cannot efficiently correct ~30% errors per DNA strand. For this reason, harnessing the diversity in DNA strands is highly advantageous, especially due to the distributive property of TdT synthesis.

We agree with the reviewer that increased sequencing bandwidth, which could be expensive, is required for obtaining the physical redundancy of multiple diverse DNA strands per template sequence. Physical redundancy has been utilized in contemporary studies for large-scale DNA storage with phosphoramidite chemistry (Organick et al. 2018). Our simulations show that for scalable DNA storage systems, only ~10 diverse DNA strands per template sequence are required for lossless reconstruction, if each DNA strand incurs less than 30% combined errors.

* The density discussion around line 266 is confusing. It seems that "redundancy" in line 271 refers to physical redundancy. And the need for higher physical redundancy is tied to any logical redundancy, which directly affects coding rate. I suggest clearly articulating coding rate, physical redundancy needs, etc.

We have revised the main manuscript to include terms for logical and physical redundancy. We clarify that an inner code for logical redundancy, which is applied to bit sequences, is different from redundancy for synchronization, which is applied to nucleotide sequences in our design (Fig. S30). Physical redundancy refers to multiple diverse DNA strands sequenced per template sequence. We have significantly revised the main manuscript throughout to include discussions about how much redundancy (i.e., synchronization, logical, and physical) is required for lossless reconstruction of data. An increase in redundancy results in a decrease in the efficiency (coding) rate of storage (i.e., the number of bits stored per template nucleotide). Similarly, a decrease in redundancy implies an increase in the efficiency

(coding) rate of storage. The efficiency (coding) rates of storage for all DNA storage systems are shown in Fig. 5A. These rates allow lossless retrieval of data from DNA.

Reviewer #2 (Remarks to the Author):

Lee et al. demonstrate a proof of concept use of a de novo enzymatic synthesis strategy that builds on the work of Palluk et al. 2018 overcomes quality and quantity limitations of the widely used phosphoramidite chemistry. They use the TdT polymerase to catalyze linkage of dNTPs with homopolymeric extensions, controlling polymerization with apyrase. Due to the repetitive nature of the homopolymer units, information is encoded in transitions between non-identical nucleotides. Synthesis costs are the primary bottleneck in making DNA data storage scalable and affordable. This approach has many advantages over phosphoramidite chemistry, particularly lower cost, shorter synthesis time, and longer template lengths.

Error-prone oligo synthesis is costly due in large part to the need to clone and sequence verify. The major shortcomings of TdT for industrial scale oligo synthesis can be tolerated in molecular data storage. To address the high error rate (primarily deletions), Lee et al. developed an error correction codec to reconstruct template sequences from incompletely synthesized strands by a scaffolding approach that uses synchronization nucleotides to locate errors. Lee et al. were able to robustly retrieve data from imperfect synthesis using a maximum a posteriori estimation of each nucleotide.

The authors demonstrated the use of nanopore sequencing over SBS. The latter is more amenable to widely used short oligo synthesis approaches, but increased template lengths will benefit from nanopore sequencing and reduction in latency, particularly without the need for single base resolution. Reaching the massive parallelization required for large-scale storage in DNA will require further improvements but Lee et al. discuss the scalability of enzymatic synthesis and demonstrate a prototype of an array-based platform.

The main bottleneck to large-scale molecular storage is the increasing number of nucleotides needed for indexing the data. The scalability of the approach to Gb/Pb is estimated from simulations, demonstrating the tradeoff between efficiency and increased error correction. Given the unmatched density of DNA data storage, this technology is will likely be applied to very large-scale data, which will require massive parallelization of synthesis, well beyond this proof-of-concept.

We agree with the reviewer that massive parallelization will be required for storing very large-scale data. Towards the goal of automation and parallelization, we have prototyped a 2D array-based platform (Figs. S31-S34) which represents a significant achievement in translation from a bead-based approach. We have demonstrated 13 cycles of synthesis for each of 3 unique DNA sequences in triplicates (S01-S03) with a 2D array prototype. Additionally, we have shown that TdT can add at least ~500 nucleotides per DNA strand in our reaction conditions (Fig. S4B). These results provide a strong basis for further development of the biochemistry and automation.

Lee et al. demonstrate considerable effort at optimizing the kinetic control of homopolymeric extension. However, a major disadvantage of phosphoramidite synthesis is the need to assemble templates > 200-300 nt. The authors cite the advantage of enzymatic synthesis to yield longer products but they do not demonstrate this important application beyond size separation of extension products. The formation of secondary structures, particularly for G homopolymers, which can contribute to the high deletion rate of TdT synthesis and loss in volumetric density warrant further consideration beyond the small scale analysis for application of in a scalable storage workflow.

We first agree with the reviewer that enzymatic synthesis has the potential to produce strands >200-300bp (Chang and Bollum 1986; Jensen and Davis 2018). This is a promising feature of enzymatic synthesis which will be a boon for directly producing >1kb strands (for genes, pathways, etc.) without requiring assembly. In this manuscript we focus on information storage, which does not require assembly, and have demonstrated that TdT can add up to ~500 nucleotides per DNA strand (Fig. S4B), consistent with previous observations about length of homopolymeric DNA synthesized by TdT (Chang and Bollum 1986). To our knowledge, the longest strand synthesized for which data is available (Palluk et al. 2018) demonstrates a sequence of 10 bases. Further automation and biochemistry development beyond the scope of this study will be required to accelerate efforts to reach maximum synthesis lengths with TdT.

We agree with the reviewer that the formation of secondary structures, such as G quadruplexes, may contribute to the rate of missing nucleotides. However, our results suggest that, under our synthesis conditions, only G homopolymers that are longer than ~10 bases inhibit further extension by TdT (Fig. S3A and S3C). Since our optimized homopolymer lengths are 3-4 bases on average, we believe that the secondary structures associated with G quadruplexes do not have a major effect on our results. Still, secondary structures of uncharacterized nature may lead to errors. To correct for these errors, a digital codec, which may rely on an inner code (logical redundancy) and/or diversity (physical redundancy), can be used but will incur a loss in volumetric density for digital information storage. We have demonstrated the scalability of the digital codec to a range of deletion rates, from succeeding in small-scale experiments (“Eureka!” experiment) to large-scale simulations with petabytes of storage capacity (Fig. S28). We have provided additional discussion in the main text about secondary structures, which can be resolved by improvements to the biochemistry or by tuning the digital codec. More experimental data beyond the scope of this manuscript will be needed to demonstrate a large-scale storage workflow.

The authors demonstrated a tolerance for high error rate but scaling up would require increased sequencing coverage, as demonstrated by the number of reads necessary for robust data retrieval of the ‘Eureka!’ encoding. Further, both homopolymer runs (>4nt) and high/low GC content show high dropout rates in PCR/sequencing.

We agree with the reviewer that increased sequencing coverage is required for our codec to resolve errors. We show in the “Eureka!” experiment that a minimum of 10 DNA strands per template sequence, each with up to 30% error (deletions, insertions, substitutions) are required for data retrieval. We observed a similar requirement in simulated larger-scale storage capacities when additional error correction is provided via an inner code (logical redundancy). Thus, the sequencing coverage required by our codec is,

in principle, as low as 10 unique strands. For this, physical isolation of the longest synthesized strands, which we did not perform, would be desirable to remove short, uninformative strands. Furthermore, we show in our simulations that improved synthesis accuracies can reduce the number of strands needed for data retrieval, and hence reduce sequencing coverage (Fig. S28).

The digital codec can be further modified to alleviate specific errors stemming from PCR and sequencing. Since information is encoded in transitions between non-identical nucleotides, alterations in extension lengths due to polymerase slippage (PCR or sequencing) in homopolymeric regions does not result in data loss. Minimizing sequences with high/low GC content may be achieved by modifying the modulation block of the codec. Further codec development and more sophisticated molecular biology techniques, such as emulsion PCR used in (Organick et al. 2018), may be required to resolve errors as storage capacities increase. In this manuscript, we resolved end-to-end errors resulting from the combination of synthesis, storage, PCR-amplification, and sequencing.

Minor points

The supplement is extremely long and there is a lot of cross reference between the main text and supplement.

We have carefully reviewed the main text and supplement for correct cross referencing. We believe that the contents of the supplement are required to sufficiently explain all aspects of this cross-disciplinary project to readers from diverse fields. The supplement also includes important details of experiments and algorithms necessary for reproducibility and further research.

Given the importance in synthesis cost reduction for scalable DNA storage, I find it strange that this is hardly discussed in the main text.

We have more explicitly pointed out the cost issues for scalable DNA storage in the introduction and have provided additional emphasis on our cost projections in the main text. Specifically, we now point out that our reagent cost analyses indicate that enzymatic synthesis can be a cheaper alternative to the phosphoramidite process with equivalent feature sizes (Fig. S35a). Additionally, further miniaturization, together with reductions to enzyme cost through recycling, provide a potential roadmap for overall reduction in reagent costs by several orders of magnitude (Fig. S35, Supplementary Text 6.2).

A nucleotide or NTP is a nucleoside triphosphate, not nucleotide triphosphate

We thank the reviewer for pointing this out and have corrected all instances in the manuscript.

References

- Blawat, Meinolf, Klaus Gaedke, Ingo Hütter, Xiao-Ming Chen, Brian Turczyk, Samuel Inverso, Benjamin W. Pruitt, and George M. Church. 2016. "Forward Error Correction for DNA Data Storage." *Procedia Computer Science* 80 (January): 1011–22.
- Chang, L. M., and F. J. Bollum. 1986. "Molecular Biology of Terminal Transferase." *CRC Critical Reviews in Biochemistry* 21 (1): 27–52.
- Church, George M., Yuan Gao, and Sriram Kosuri. 2012. "Next-Generation Digital Information Storage in DNA." *Science* 337 (6102): 1628.
- Erlich, Yaniv, and Dina Zielinski. 2017. "DNA Fountain Enables a Robust and Efficient Storage Architecture." *Science* 355 (6328): 950–54.
- Goldman, Nick, Paul Bertone, Siyuan Chen, Christophe Dessimoz, Emily M. LeProust, Botond Sipos, and Ewan Birney. 2013. "Towards Practical, High-Capacity, Low-Maintenance Information Storage in Synthesized DNA." *Nature* 494 (7435): 77–80.
- Jensen, Michael A., and Ronald W. Davis. 2018. "Template-Independent Enzymatic Oligonucleotide Synthesis (TiEOS): Its History, Prospects, and Challenges." *Biochemistry* 57 (12): 1821–32.
- Organick, Lee, Siena Dumas Ang, Yuan-Jyue Chen, Randolph Lopez, Sergey Yekhanin, Konstantin Makarychev, Miklos Z. Racz, et al. 2018. "Random Access in Large-Scale DNA Data Storage." *Nature Biotechnology* 36 (3): 242–48.
- Palluk, Sebastian, Daniel H. Arlow, Tristan de Rond, Sebastian Barthel, Justine S. Kang, Rathin Bector, Hratch M. Baghdassarian, et al. 2018. "De Novo DNA Synthesis Using Polymerase-Nucleotide Conjugates." *Nature Biotechnology* 36 (7): 645–50.

Reviewers' Comments:

Reviewer #1:

Remarks to the Author:

Thank you for sharing your response to the review and addressing some of the concerns on the revised manuscript.

Reviewer #2:

Remarks to the Author:

All were addressed in detail in the response to reviewers and incorporated in the main text. I have no additional comments.

Point-by-point Response (Reviewers)

Please view all original, unmodified comments by reviewers in black. Similarly, please view our responses to all questions highlighted in blue.

Reviewer #1 (Remarks to the Author):

Thank you for sharing your response to the review and addressing some of the concerns on the revised manuscript.

We thank the reviewers for their insightful comments towards strengthening the manuscript.

Reviewer #2 (Remarks to the Author):

All were addressed in detail in the response to reviewers and incorporated in the main text. I have no additional comments.

We thank the reviewers for their insightful comments towards strengthening the manuscript.